# Qualitative and Quantitative Differences in Osmolytes Accumulation and Antioxidant Activities in Response to Water Deficit in Four Mediterranean *Limonium* Species

**DOI:** 10.3390/plants8110506

**Published:** 2019-11-15

**Authors:** Sara González-Orenga, Mohamad Al Hassan, Josep V. Llinares, Purificación Lisón, M. Pilar López-Gresa, Mercedes Verdeguer, Oscar Vicente, Monica Boscaiu

**Affiliations:** 1Instituto Agroforestal Mediterráneo (IAM), Universitat Politècnica de València, Camino de Vera s/n, 46022 Valencia, Spain; sagonor@doctor.upv.es (S.G.-O.); jollipa@qim.upv.es (J.V.L.); merversa@doctor.upv.es (M.V.); 2Instituto de Biología Molecular y Celular de Plantas (IBMCP), Universitat Politècnica de València - Consejo Superior de Investigaciones Científicas (CSIC), Camino de Vera s/n, 46022 Valencia, Spain; mohamed.alhassan@wur.nl (M.A.H.); plison@ibmcp.upv.es (P.L.); mplopez@ceqa.upv.es (M.P.L.-G.); 3Wageningen UR Plant Breeding, Wageningen University and Research Centre, Droevendaalsesteeg 4, 6708 PB Wageningen, The Netherlands; 4Instituto de Conservación y Mejora de la Agrodiversidad Valenciana (COMAV), Universitat Politècnica de València, Camino de Vera s/n, 46022 Valencia, Spain; ovicente@upvnet.upv.es

**Keywords:** *Limonium santapolense*, *Limonium virgatum*, *Limonium girardianum*, *Limonium narbonense*, drought, water deficit, oxidative stress, ions, osmolytes, antioxidant enzymes

## Abstract

*Limonium* is a genus represented in the Iberian Peninsula by numerous halophytic species that are affected in nature by salinity, and often by prolonged drought episodes. Responses to water deficit have been studied in four Mediterranean *Limonium* species, previously investigated regarding salt tolerance mechanisms. The levels of biochemical markers, associated with specific responses—photosynthetic pigments, mono- and divalent ions, osmolytes, antioxidant compounds and enzymes—were determined in the control and water-stressed plants, and correlated with their relative degree of stress-induced growth inhibition. All the tested *Limonium* taxa are relatively resistant to drought on the basis of both the constitutive presence of high leaf ion levels that contribute to osmotic adjustment, and the stress-induced accumulation of osmolytes and increased activity of antioxidant enzymes, albeit with different qualitative and quantitative induction patterns. *Limonium santapolense* activated the strongest responses and clearly differed from *Limonium virgatum*, *Limonium girardianum,* and *Limonium narbonense,* as indicated by cluster and principal component analysis (PCA) analyses in agreement with its drier natural habitat, and compared to that of the other plants. Somewhat surprisingly, however, *L. santapolense* was the species most affected by water deficit in growth inhibition terms, which suggests the existence of additional mechanisms of defense operating in the field that cannot be mimicked in greenhouses.

## 1. Introduction

In general, Mediterranean plants are well adapted to drought, marked by drastic reduction in summer rainfall and wide inter-annual variability, both of which characterize the Mediterranean climate [1]. Nevertheless, forecasts estimate that environmental conditions will become more stressful from global warming, particularly in the Mediterranean region, and that droughts will become severer and more frequent [2].

Drought induces water deficit in stressed plants, and this strongly impacts all plant organs to a greater or lesser extent [3]. This shortage of water brings about a range of deleterious effects, from reactive oxygen species (ROS) levels building up and producing oxidative stress [4], to a low photosynthesis potential [5]. Almost every aspect of drought-affected plant homeostasis is negatively altered, which implies reduced vegetative growth, yield, and eventually plant death [6].

However, plant roots resort to effective mechanisms that sense low water potential. This scenario may emerge due to lack of water in the environment caused by low precipitation or excess salt ions being present in soil, which leads to ‘physiological drought’ [7]. In both events, plants are unable to take up enough water for normal development and growth, activating stress-related signal transduction pathways [8], which is immediately followed by stunted shoot and leaf growth. This inhibited growth is linked with a change in carbon dioxide and cellular oxygen levels prompted by partial stomata closure [3].

Nevertheless, plants possess several tolerance mechanisms to alleviate effects of drought-induced osmotic stress, including the synthesis of compatible solutes, for example, proline, glycine betaine, or soluble carbohydrates [9,10,11,12]. Osmotic adjustment is also accomplished by inorganic ions accumulating. High K^+^ and Na^+^ levels in water-stressed plants have been described in the stress-tolerant *Atriplex halimus* [10], whereas Na^+^ accumulation has been found to act as an important drought tolerance strategy in a desert xerophyte [11].

Osmotic stress is linked with increased reactive oxygen species (ROS) production. Drought, thus, comes with raised ROS levels, which brings about changes in cellular redox homeostasis and normal cellular metabolism that trigger oxidative stress and the activation of antioxidant mechanisms [13,14,15]. The most common antioxidant metabolites include phenols, flavonoids, ascorbic acid, glutathione, and carotenoids, whereas catalase (CAT), superoxide dismutase (SOD), ascorbate peroxidase (APX) (and other peroxidases), or redox regulatory enzymes such as glutathione reductase (GR), are among the most relevant antioxidant enzymatic systems activated in plants to respond to deleterious oxidative stress effects [13].

The genus *Limonium* L. of the Plumbaginaceae family comprises over 400 species, includes many endemic species in the present study area, and is well represented in the Mediterranean region [16]. The *Limonium* species have been well documented for responding to salt stress because this is an emblematic genus of halophytes that possesses salt excretory glands [17,18]. However, the mechanisms of response and potential tolerance to drought of *Limonium* taxa, similarly to the majority of halophytes, have not drawn much interest and are still largely unknown. Nonetheless, salinity is not the only constraint for plants in salt marshes because, quite often, many other stressful factors occur simultaneously.

In this and former works, we examine the responses of four *Limonium* species to abiotic stress: *Limonium santapolense* Erben, *Limonium girardianum* (Guss.) Fourr., *Limonium virgatum* (Willd.) Fourr., and *Limonium narbonense* Mill. All four are found in littoral salt marshes in southeastern Spain. They are perennial, long day, and C3, and their leaves differ in size and shape. The first two are endemics and have a high conservation value. *L. santapolense* is found on littoral sandy substrates in a small zone in the Province of Alicante. *L. girardianum* is endemic to eastern Spain, southern France, and the Balearic Isles, growing on cliffs and sandy coasts. The other two species are broadly distributed and cover the Mediterranean region, with *L. virgatum* found on rocky coasts and sandy beaches reaching North Africa and the Middle East, whereas *L. narbonense* is present in salt marshes throughout the Mediterranean, including Spain, as well as on the Atlantic coast [16].

The objective of this work is to analyze the responses of all four *Limonium* species to water stress generated artificially and under greenhouse conditions to plants grown from field-collected seeds. We hypothesize that the drought tolerance mechanisms of *L. santapolense* plants, whose origin lies in the aridest collection area, would be more efficient than those of the plants of the other three chosen species. In our extensive study, growth inhibition, induced by not watering plants for 1 month, correlated with (a) degradation of photosynthetic pigments; (b) ionic homeostasis being maintained; (c) compatible solutes accumulating, for example, proline, glycine betaine, and soluble carbohydrates; (d) leaf malondialdehyde (MDA, a reliable oxidative stress marker) and H_2_O_2_ contents, and α,α-diphenyl-β-picrylhydrazyl (DPPH) free radical scavenging activity; (e) antioxidant compound (total phenolics and flavonoids) levels; and (f) antioxidant enzymes activities (CAT, GR, SOD, and APX).

## 2. Results

### 2.1. Substrate Moisture at the End of Treatments

After not irrigating the four analyzed species for 1 month, substrate humidity was significantly lower than that recorded in the control treatments, in which plants were watered twice weekly (Table 1). When comparing the four species, no significant differences were found in the moisture values of substrates (Table 1).

### 2.2. Effects of Drought on Plant Growth and Photosynthetic Pigments Levels

Water stress caused significantly inhibited vegetative plant growth in two of the selected species, *L. santapolense* and *L. narbonense*, as shown by the reduced fresh weight (FW) of aerial plant parts by about one third versus the corresponding controls (Figure 1, Table 2). Slight reduction (approximately 11%) was recorded in *L. girardianum*, whereas the mean FW of the water-stressed *L. virgatum* plants was slightly higher than in the control. However, the differences found for the two latter species were not significant (Table 2).

Regarding the reduced leaf area (LA), the effect of drought was statistically significant only in *L. santapolense*, the species with broader leaves, which lost about 40% of its foliar area compared to the non-stressed control. Despite complete withholding irrigation for 1 month, the dehydration of the *Limonium* plants was very low. Once again, the drought-induced reduction of the leaf water content (WCL) was significant only in *L. santapolense* and came to less than 2%. On the contrary, the root water content (WCR) reduction was significant only in *L. narbonense* (Table 2). According to these growth parameters, *L. santapolense* can be considered the species most affected by water stress under our experimental conditions.

Water stress induced only minor, non-significant changes in the leaf contents of photosynthetic pigments (chlorophylls a and b, and total carotenoids) in the four *Limonium* species (Table 2). Pigment levels were similar in all four species, except for chlorophyll a in *L. virgatum* and *L. girardianum*, which showed a significant difference when compared to the control (untreated) plants of the two species.

### 2.3. Ion Accumulation

No changes in the leaf contents of the mono- (Na^+^, K^+^, and Cl^−^) and divalent (Ca^2+^ and Mg^2+^) ions were induced under the water deficit conditions in any of the investigated *Limonium* species (Table 3). Some significant differences between the control and stressed plants were, however, observed in roots, but with no clear pattern of variation. For example, Na^+^ and Cl^−^ generally increased in response to water stress, although the differences with the corresponding controls were not significant in all the *Limonium* taxa, whereas K^+^ contents did not vary. The mean Ca^2+^ and Mg^2+^ levels drastically dropped in *L. santapolense,* but rose in the other three species (Table 3). The K^+^/Na^+^ ratio, which is considered relevant for maintaining ionic homeostasis, did not vary in the plant leaves, but significantly decreased in the roots of all the species, except *L. virgatum*, which showed a non-significant reduction. When comparing ion contents in the roots and leaves of the same plants, they were all higher in leaves in both the control and water-stressed plants—two- to fourfold, approximately, for the monovalent ions in all four *Limonium* species, and for Ca^2+^ and Mg^2+^ in *L. narbonense*. For the other three species, the divalent cation levels were about 6- to 15-fold higher in leaves than in roots (Table 3). The relatively high Mg^2+^ concentrations measured in the leaves of these species is also worth mentioning.

### 2.4. Water Stress-Induced Osmolyte Accumulation

The levels of the commonest plant osmolytes—proline (Pro), glycine betaine (GB), and total soluble sugars (TSS)—were determined in the leaves of the investigated *Limonium* taxa when the water deficit treatments ended (Figure 2). With the exception of *L. narbonense*, Pro levels increased significantly compared to the untreated controls. However, this stress-induced increment was by far more pronounced in *L. santapolense*, where Pro reached 122 µmol g^−1^ DW, which represented a sixfold increase over the control (Figure 2a).

The leaf GB contents in the control plants were similar to those of Pro (20–50 µmol g^−1^ DW), except for *L. virgatum*, in which a concentration of 76 µmol g^−1^ DW was determined. These values only slightly varied in general, and not significantly, in the plants subjected to the water deficit treatment (Figure 2b).

*Limonium santapolense* showed higher leaf TSS contents than the other three species in the control plants. For all four taxa, the water stress treatment induced only minimal changes in the TSS levels, as observed for GB (Figure 2c).

To check the possibility that the leaf levels of particular sugars could vary in response to the water deficit treatment, which cannot be detected by measuring TSS contents, soluble carbohydrates in aqueous extracts of plants were separated, identified, and quantified by HPLC (Figure 3). Three major peaks in the chromatograms were observed, corresponding to glucose (Glu), fructose (Fru), and sucrose (Suc), but their concentrations showed entirely different patterns in the four analyzed *Limonium* species. Glu was detected only in *L. giradianum* and *L. narbonense*, but its concentration increased significantly in response to drought only in the former species by reaching 15.5 µmol g^−1^ DW (Figure 3a). In *L. santapolense* and *L. virgatum*, glucose levels were below the detection limit of the evaporative light scattering detector (ELSD), both in control and stressed plants. Fru concentrations were very low (<0.5 µmol g^−1^ DW) in the non-stressed *L. santapolense* plants, and increased considerably to ca. 38 µmol g^−1^ DW in response to water stress. The other three species had higher control Fru values (8–17 µmol g^−1^ DW), which either lowered or did not change significantly in the stressed plants (Figure 3b). Large differences in leaf Suc contents were also detected in the control plants of the four analyzed species, which went from extremely low values (<0.2 µmol g^−1^ DW) in *L. giradianum* to 2.2–2.4 µmol g^−1^ DW in *L. narbonense* and *L. virgatum*, and to ~17 µmol g^−1^ DW in *L. santapolense*. In the latter species, water deficit stress induced a significant increase of 1.4-fold in the Suc concentration (Figure 3c), which was much lower than that observed for Fru in any case (Figure 3b).

### 2.5. Oxidative Stress and Activation of Antioxidant Systems

Malondialdehyde (MDA) levels did not significantly differ in the control plants of the four *Limonium* species. After the water stress treatment, leaf MDA content increased in all cases, albeit slightly, between 1.1- and 1.7-fold depending on the species. However, the differences with the non-stressed controls were statistically significant only in *L. narbonense* (Table 4). Hydrogen peroxide levels were similar in the control plants of all the *Limonium* taxa, and the water deficit-induced variations were also non-significant (Table 4). The overall antioxidant activity of plant extracts, as determined by the DPPH (α,α-diphenyl-β-picrylhydrazyl) free radical scavenging assay, did not change in the stressed plants compared to their corresponding controls. In this case, however, significant differences between taxa were observed, with *L. santapolense* showing the greatest antioxidant activity (Table 4).

Phenolic compounds, especially, the subclass of flavonoids, are well-established examples of antioxidant metabolites. Leaf levels of total phenolic compounds (TPC) and total flavonoids (TF) were determined in the control and water-stressed plants. Once again, TPC contents were higher in *L. santapolense* than in the other three species, but did not change as a result of the water deficit treatment in all cases. TF levels, however, were low and did not vary significantly either between species or between the control and stressed plants (Table 4). The leaf levels of anthocyanins, which can also be used to estimate the antioxidant activity of plant samples, were low in all the control plants and did not vary significantly in response to the water deficit treatment (data not shown).

The specific activities of the four tested antioxidant enzymes (SOD, CAT, APX, and GR) showed different patterns in the four species for both basal values in the controls and quantitative changes in response to water deficit. In most cases, however, the stress treatment led to greater antioxidant activities (Figure 4). SOD activity, for example, increased significantly in *L. santapolense* and *L. giradianum*, but did not change in *L. virgatum* and *L. narbonense*. The highest specific activity was measured in the stressed *L. santapolense* plants (Figure 4a). CAT activity was very low in the control *L. santapolense* and *L. virgatum* plants and slightly but significantly augmented in response to the drought treatment. The control values were much higher in *L. narbonense* and also grew under stress conditions, but did not vary in *L. giradianum* (Figure 4b). APX activity increased significantly in the four species, but reached much higher values in *L. santapolense* than in the other three taxa (Figure 4c). GR activity increased in response to the stress treatment in all the *Limonium* taxa, except for *L. giradianum*. Once again, the greatest activity was measured in *L. santapolense,* but the relative increment over the control values was lower than in *L. virgatum* and *L. giradianum* (Figure 4d).

### 2.6. Statistical Analysis of Data: Factorial ANOVA, Clustering of Species, and Principal Component Analysis

The results of a factorial ANOVA considering the effect of treatment, species, and their interaction are shown in Table 5. Of the 31 parameters analyzed, 27 varied significantly according to the species, but only 15 according to the treatment. The greatest variations between values measured in control and water-stressed plants were registered for osmolytes and antioxidant enzymes. Interestingly, the interactions between the two factors (treatment × species) were also significant for these parameters, indicating that the tested species do not show the same patterns of response to water stress.

The cluster analysis performed using all the variables measured in the water-stressed plants (including growth parameters) clearly distinguished *L. santapolense* individuals from the others as they were present on a separate branch of the dendrogram (Figure 5). Of the remaining three species, *L. narbonense* was the most distant, whereas the values recorded in *L. virgatum* and *L. girardianum* were entangled. A principal component analysis (PCA) was also performed, including the growth parameters, osmolytes, MDA, and enzyme activities determined in the water stress and control treatments. Photosynthetic pigments, ions, H_2_O_2_, and antioxidant compounds were not included as they did not vary significantly under stress. Ten components had an eigenvalue above 1. The biplot of the two main principal components, which together explained 63% of total variability, is shown in Figure 6. The first component, which explains the highest variability of the data (41.79%) was related mostly to the treatment, whereas the second (20.18%) was related to the species. Water stress, reflected as reduced substrate moisture after not watering pots for 1 month, correlated positively with changes in water content in the plant roots and leaves, and negatively with proline and fructose (the osmolytes that strongly increased under stress), and with antioxidant enzymes (especially GR and APX), whose activity also increased as a response to water deficit. The PCA, like the cluster analysis, showed a clear separation of *L. santapolense* from the other taxa, which all appeared together in the negative sector of the *x*-axis. *L. santapolense* was the only species for which the control and stress treatment values were clearly distant. The latter correlated positively with changes in the Pro and Fru levels, and in the activities of SOD, APX, and GR, which thus confirms the results of the above-described individual experiments (Figure 2, Figure 3 and Figure 4).

## 3. Discussion

A considerable number of relevant physiological studies has been recently published on Mediterranean plant species that are adapted to severe stressful environments. Several have specifically centered on distinct *Limonium* species, or have included some taxa of this genus along with other species [19,20]. With regards to the activation of specific stress responses at molecular and biochemical levels, the majority of former research works into *Limonium* have centered on responses to salt stress [17,18]. All this renders novel the data herein presented, and stresses the role played by antioxidant enzymes and specific osmolytes in mechanisms of defense against water deficit of *Limonium* plants.

Of all four studied species, *L. santapolense* was the most strongly affected in the greenhouse experiments by water deficit stress, as indicated by its severely inhibited growth (reduced mean leaf area and fresh weight vs. controls) under stress conditions. The only taxon to display a significant, albeit low, degree of leaf dehydration in the unwatered plants was *Limonium santapolense*. Our experiments revealed how the other three species tolerated water stress quite well. Of these, a significantly reduced fresh weight in the stressed plants was evidenced only for *L. narbonense*. In these two species, the apparently diminished water deficit tolerance could be associated with the morphological characteristics of all four taxa, specifically with their leaf size because their leaves are larger and broader than those of *L. girardianum* and *L. virgatum*. This particular trait is generally thought to enhance plant sensitivity to dehydration, and there are reports that plants can respond to water deficit caused by the global warming by them diminishing their leaf areas [21].

Water stress had no effect on the levels of photosynthetic pigments in any studied *Limonium* taxon, including *L. santapolense*. As inhibited photosynthesis and low chlorophyll contents are some frequent effects of drought and other abiotic stresses [22,23], these results also indicate that the studied species are quite resistant to water stress.

In the experiments performed herein, plant growth took place with low salt concentrations, those present in the nutrient solution and the peat substrate, and no major changes were expected in the plant ion contents in response to water stress. However, as inorganic ions can contribute to cellular osmotic adjustment with drought [24], the levels of the mono- and divalent cations, and those of Cl^−^, were determined in the leaves and roots of the stressed and control plants. In roots, save *L. virgatum*, Cl^−^ and Na^+^ content significantly rose in the plants undergoing the water stress treatment versus the corresponding controls. This can be accounted for by the activation of ion transport in plants by counteracting, at least in part, the osmotic stress produced by not watering the plants. Increasing Na^+^ concentrations are usually accompanied by loss of K^+^, as Na^+^ interferes with K^+^ uptake by employing the same transport systems, and both cations compete for the same binding proteins [3]. These changes in K^+^ and Na^+^ contents give rise to lower K^+^/Na^+^ ratios. Indeed, we recorded this reduction in the root K^+^/Na^+^ ratios in the *Limonium* plants (save *L. virgatum*), despite no significant drop in the K^+^ concentration being detected. Ion leaf contents displayed a distinct pattern. Firstly, and most importantly, the levels of the three determined monovalent ions (K^+^, Na^+^, and Cl^−^), and those of divalent cations Mg^2+^ and Ca^2+^, were significantly higher in leaves than in roots. This finding clearly suggests the presence of active transport systems of these ions to aerial plant parts, which was also found under salt stress in *Limonium* [18]. High leaf concentrations of K^+^ and Na^+^ also play an important role in osmotic adjustment under water stress in quinoa, as has been recently reported [25]. Secondly, significant differences were not found between the stressed and control plants in ion leaf contents for any ions or any species. This would indicate that ion transport activation is not induced by water deficit, but is likely a constitutive mechanism of response to stress that *Limonium* plants use to help contribute to the cellular osmotic balance in leaves.

As for the biochemical responses of all four *Limonium* taxa, *L. santapolense* behaved differently according to the cluster analysis, which showed a clear separation between this species and the remaining three. The most striking differences to appear among them indicate the activation of different antioxidant enzymes and the stress-induced accumulation of particular osmolytes. Water deficit led to the considerable accumulation of both fructose and proline in *L. santapolense* leaves, at concentrations around 40 and 120 µmol g^−1^ DW, respectively, which were much higher than those recorded in the controls. Sucrose levels also significantly increased, but by less than 1.5-fold, with lower absolute values (<25 µmol g^−1^ DW). We conclude that osmotic adjustment to protect plants from dehydration in *L. santapolense* under water stress conditions is based on Fru and Pro accumulation, with Suc contributing less. The other *Limonium* taxa exhibited a much weaker water stress response for the mechanisms mediated by accumulating compatible solutes. In the three species, the leaf contents of all the tested putative osmolytes slightly increased, or not at all, as a response to stress treatment, except for glucose accumulation in *L. girardianum,* whereas the absolute concentrations of these compounds were too low to have a strong osmotic effect.

The lack of glucose accumulation at detectable levels in *L. santapolense* and *L. virgatum*, both in unstressed plants and in those subjected to water deficit, should be mentioned. Similarly, very low, in most cases non-detectable, glucose leaf contents were determined in a previous study, in control and salt-stressed plants of the same taxa [18]. These data suggest that glucose does not play any relevant role in osmotic adjustment under stress in these particular species. The molecular basis of this behavior is not known, but could be possibly related to a rapid turnover of glucose in these taxa, used as an energy source and/or as a building block for the synthesis of polysaccharides required under stress conditions. For example, it has been recently reported that drought causes changes in the cell walls in leaves and stems of miscanthus, a biofuel crop, with an increase in their hemicellulose contents [26]. In any case, *Limonium* taxa are characterized by a large diversity in the type of compatible solutes that accumulate in response to abiotic stress treatments. The osmolytes herein identified have already been reported in other *Limonium* species, including plant material from natural habitats [18,27], but along with a considerable number of other compounds, such as choline-O-sulfate, alanine, betaine, or distinct polyalcohols (e.g., *myo*-inositol, pinitol, or *chiro*-inositol) [17,28,29]. The simultaneous synthesis of distinct osmolytes has been observed in *Limonium* [17] as in other Plumbaginaceae species [30]. The concomitant synthesis of distinct osmolytes is a helpful strategy adopted by stress-tolerant taxa because it enables them to adapt better to the stressful environments they live in [31].

Drought, just like any other abiotic stress, increases levels of ROS, which, in excess, oxidize unsaturated fatty acids in cell membranes, amino acid residues in proteins, and DNA molecules, and thus provoke cellular damage [32]. Hydrogen peroxide is the most relevant stable non radical among ROS that is produced in peroxisomes and chloroplasts, and is thought to be a good marker of the extent of oxidative stress [33]. According to our experiments, the leaf H_2_O_2_ levels showed no marked variation in the stressed plants versus the non-stressed controls in the studied four species. Malondialdehyde (MDA) is a lipid peroxidation product employed as a reliable oxidative stress marker in both animals and plants [34]. The standard method followed to assess the ability of compounds to act as hydrogen donors or free radical scavengers is DPPH free radical scavenging by indicating specific biological samples’ general antioxidant activity [35]. For all four *Limonium* taxa herein analyzed, neither the total free radical scavenging activity of the leaf extracts nor MDA contents significantly differed between the water-stressed and control plants, except for a slight, but statistically significant, increase in MDA levels in the leaves of *L. narbonense*. These findings indicate that the water deficit treatment did not lead to a detectable degree of oxidative stress in the plants, which was likely owing to the activation of efficient antioxidant systems. However, such systems do not include antioxidant compounds such as flavonoids or phenolic compounds, in general, because their leaf concentrations did not vary in response to stress treatment. We found similar results in halophytes sampled in the wild, which did not show a seasonal variation of MDA, although environmental conditions drastically changed in summer [36]. A review on ROS homeostasis in halophytes has demonstrated that they do not require high antioxidant activity levels because they do not generate ROS in excess thanks to their efficient mechanisms that avoid oxidative stress [37]. It is believed that enzymatic antioxidant systems constitute the first line of defense against oxidative stress, whereas phenolic compounds (including flavonoids and other metabolites with antioxidant activity) are a secondary ROS scavenging system that is activated only under severe stress conditions when antioxidant enzymes do not suffice [38]. Thus, with the *Limonium* species chosen for this research, activation of antioxidant enzymes seemed sufficient to counteract the oxidative stress that the water deficit treatment generated, as formerly reported for other *Limonium* species under salt stress conditions [39].

SOD constitutes primary defense against ROS by catalyzing dismutation of superoxide radicals into O_2_ and H_2_O_2_ [40]. SOD-specific activity is enhanced when the superoxide substrate is present by the transcriptional activation of corresponding genes, that is, by the de novo synthesis of the enzyme [41]. After SOD, CAT acts by decomposing the produced H_2_O_2_ into O_2_ and H_2_O, induced by its substrate accumulating [42]. APX catalyzes the reduction of H_2_O_2_, coupled to ascorbate oxidation. GR contributes to recover and maintain the adequate cellular redox state by reducing oxidized glutathione (GSSG) to its reduced form (GSH), which it does by employing NADPH as a cofactor [43].

In general, the specific activities of these antioxidant enzymes were enhanced in response to water stress in all four selected *Limonium* species, albeit quantitative differences were found in different taxa. *Limonium santapolense* displayed the strongest response as the four activities significantly increased in the stressed plants versus the controls. The induced SOD, APX, and GR levels were higher than those of the other three species, and only CAT displayed less activity, which was below that measured in *L. narbonense* or *L. girardianum*. Different patterns were observed for enzymatic antioxidant responses in the remaining taxa. SOD activity in *L. virgatum* did not vary in the water-stressed plants and remained quite high in the non-stressed controls. APX, CAT, and GR activities significantly increased with stress, but the absolute activity values remained very low for APX and CAT. In *L. girardianum,* antioxidant defense was dependent on the constitutive presence of relatively strong GR and CAT activities, with water stress-induced SOD and APX contributing less. Finally, water stress in *L. narbonense* did not activate SOD, but instead induced marked increases in GR and CAT activities, and also in APX to a much lesser extent.

In summary, the four analyzed *Limonium* taxa are relatively resistant to drought, partly on the basis of the presence of constitutive stress tolerance mechanisms, such as the active transport of mono an divalent cations to the leaves, contributing to osmotic balance, as well as on the water deficit-induced accumulation of specific osmolytes and the increased activity of antioxidant enzymes. These induced responses showed different qualitative and quantitative patterns, allowing a clear separation of *L. santapolense* from the other three taxa; this species activated the strongest drought responses through the specific accumulation of high levels of Pro and Fru, as functional osmolytes, and the significant increase in the specific activities of the assayed antioxidant enzyme systems, SOD, APX, GR, and, to a lesser extent, CAT.

## 4. Materials and Methods

### 4.1. Sampling Sites and Seed Sampling

Mature capsules of *L. santapolense* were collected from Clot de Galvany, a salt marsh located near the city of Elche in the Province of Alicante (39.12° N/0.20° E), and those of the other three species came from the ‘La Albufera’ Natural Park near the city of Valencia (38°15 N/0.42° W), Spain, in autumn 2016. Seeds were separated from capsules and stored at room temperature for 2 months.

### 4.2. Plant Growth, Drought Treatments, and Sampling of Plant Material

Plants were obtained by directly sowing seeds on a mixture of commercial peat and vermiculite (3:1). Seedlings were watered twice weekly with Hoagland nutrient solution [44]. After 3 weeks, seedlings were transferred individually to 1 L pots and placed in plastic trays (five pots per tray). One week later, stress treatments were initiated by entirely ceasing irrigation. The plants from the control treatment were watered every 5 days with 1 L water added to each tray. After 1 month of treatment, five stressed plants of each species and their control counterparts were harvested, together with a fraction of the corresponding substrates. All the experiments were conducted in a controlled environment chamber in a greenhouse under the following conditions: long-day photoperiod (16 h of light), temperature set at 20 °C during the day and 17 °C at night, and relative humidity between 50% and 80%, monitored by a Testo humidity data logger.

The leaves and roots of each harvested plant were collected and separately weighed. The following plant growth parameters were measured in the leaf fraction: fresh weight of leaves (FWL), leaf water content percentage (WCL), and leaf surface (LA). Five leaves from each plant were selected randomly and scanned to measure the leaf surface with the ImageJ software [45]. A fraction of the fresh material was frozen in liquid N_2_ and stored at −75 °C. Most of the remaining material was dried for several days in an oven at 65 °C until constant weight was achieved. The water content percentage in leaves was calculated as WCL (%) = [(FWL − DWL)/FWL] × 100 [36]. The root fraction of each harvested plant was thoroughly cleaned by brushing with a fine paintbrush. Then, roots were briefly rinsed in Milli-Q water, quickly blotted on filter paper, and dried at 65 °C to calculate the water content percentage of roots (WCL), as for the leaf fraction. The total FW of roots could not be determined because it was not possible to recover the whole root system of each plant.

### 4.3. Substrate Analysis

Soil moisture was determined by the gravimetric method at the end of treatments—a fraction of each soil sample was weighed (SFW), dried in an oven at 105 °C until reaching constant weight, and then weighed again (DSW). Soil water content (in %) was calculated as

Soil humidity: WC% = [(FSW − DSW)/FSW] × 100.(1)

### 4.4. Photosynthetic Pigments

Chlorophylls a and b (Chl a, Chl b) and total carotenoids (Caro) were determined as previously described [46]. Ten mL of ice-cold 80% (*v*/*v*) acetone was used to extract pigments from 0.05 g of fresh leaf material. After mixing overnight and centrifuging for 10 min at 12,000 rpm, the supernatant was collected, and its absorbance was measured at 663, 646, and 470 nm. Chl a, Chl b, and Caro concentrations were calculated using described equations [46] and their contents were expressed in mg g^−1^ DW.

### 4.5. Ion Content Measurements

Ion contents were determined in root and leaf aqueous extracts, essentially as described by Weimberg [47], by heating samples (0.05 g of dried ground plant material in 15 mL of water) for 15 min at 99 °C, followed by filtration through a 0.45 µm filter (Gelman Laboratory, PALL Corporation, Port Washington, NY, USA. Na^+^ and K^+^ were quantified with a PFP7 flame photometer (Jenway Inc., Burlington, VT, USA). Cl^−^ was measured using a chloride analyzer. Divalent cations (Ca^2+^ and Mg^2+^) were determined in an atomic absorption spectrometer SpectrAA 220 (Varian, Inc., Palo Alto, CA, USA).

### 4.6. Osmolyte Quantification

Proline (Pro) was extracted from 0.05 g of leaf fresh material with 2 mL of a 3% (*w*/*v*) sulfosalicylic acid solution to be quantified according to the acid ninhydrin method [48]. The extract, mixed with acid ninhydrin, was heated at 95 °C for 1 h, cooled on ice, and extracted with toluene. The absorbance of the organic phase was measured at 520 nm using toluene as a blank. Pro concentrations were expressed as μmol g^−1^ DW. Glycine betaine (GB) was determined in 1-mL aqueous extracts prepared from 0.05 g of the dry leaf material according to published procedures [49,50]. The extract was supplemented with potassium iodide, kept on ice for 90 min, and then extracted with 1,2-dichlorethane (pre-cooled at −20 °C). Finally, the absorbance of the sample was measured at 365 nm. GB content was expressed as μmol g^−1^ DW. To quantify total soluble sugars (TSS), 0.05 g of the dry ground leaf material was extracted with 3 mL of 80% (*v*/*v*) methanol and mixed on a rocker shaker for 24 h. The extract was centrifuged, concentrated sulfuric acid and 5% phenol were added to the supernatant, and absorbance was measured at 490 nm [51]. TSS contents were expressed as ‘milligram equivalents of glucose’ (used as the standard) per g DW.

### 4.7. HPLC Analysis of Soluble Carbohydrates

Plant fresh material (0.05 g) was boiled in 2 mL of milliQ water for 10 min before being filtered using 0.22 µm filters. The soluble sugar fraction was analyzed using a Waters 1525 HPLC system coupled to a 2424 evaporative light scattering detector (ELSD), as previously described [52]. The source parameters of ELSD were the following: gain 75, data rate one point per second, nebulizer heating 60%, drift tube 50 °C, and gas pressure 2.8 Kg/cm^2^. Samples of 20 µL were injected into a Prontosil 120-3-amino column (4.6 × 125 mm; 3 µm particle size) and were maintained at room temperature with a Waters 717 auto-sampler. An isocratic flux (1 mL/min) of 85% acetonitrile (J.T. Baker) was applied in each run for 25 min. Glucose, fructose, and sucrose standards were employed to identify peaks by co-injection. Sugars were quantified with peak integration using the Waters Empower software, and comparisons were made with the glucose, fructose, and sucrose standard calibration curves.

### 4.8. MDA, H_2_O_2_, DPPH, and Non-Enzymatic Antioxidants

MDA contents were determined essentially as described [53], with modifications [54]. Methanol extracts (80% *v*/*v*) of the leaf material were mixed with 0.5% thiobarbituric acid (TBA) in 20% TCA, and then incubated 15 min at 95 °C. The reaction was stopped on ice; the absorbance of the sample was measured at 440, 600, and 532 nm; and the MDA concentration was determined using the described equations [54].

The leaf hydrogen peroxide contents in both the control and treated plants were quantified as previously described [55], with minor modifications. Dried leaf material (0.05 g) was extracted with a 0.1% (*w*/*v*) trichloroacetic acid (TCA) solution, followed by centrifuging the extract. The supernatant was thoroughly mixed with one volume of 10 mM potassium phosphate buffer (pH 7) and two volumes of 1 M potassium iodide. The absorbance of the sample was determined at 390 nm. Hydrogen peroxide concentrations were calculated against an H_2_O_2_ standard calibration curve and expressed as µmol g^−1^ DW.

The total antioxidant activity of extracts was evaluated by measuring the ability of samples to quench the stable radical 2,2-diphenyl-1-picrylhydrazyl (DPPH), a synthetic free radical product whose quenching by a scavenger substrate can be followed spectrophotometrically at 517 nm [35]. Plant dry material (0.05 g) was extracted in 2 mL of 90% methanol, sonicated for 10 min, and centrifuged at 14,000 rpm for 15 min. Then, 50 μL of the soluble fraction was diluted with 2 mL of 96% ethanol. A fraction of the resulting solution was diluted four times with 96% ethanol containing 125 µM DPPH. The reaction mixture was incubated at 25 °C for 10 min, and the absorbance of the sample was measured at 517 nm. A blank sample with no plant extract was included to check radical stability. The radical scavenging activity (*S*) of each extract was expressed as a percentage and calculated as
*S* = 100 − [(*A_x_*/*A*_0_) × 100](2)
where *A_x_* is the absorbance of the DPPH solution in the presence of the extract, and *A*_0_ is the absorbance of the blank.

Total phenolic compounds (TPC) and total flavonoid (TF) contents were determined in the same methanol extracts used for the TSS measurements. TPC were quantified by running a reaction with the Folin–Ciocalteu reagent [56]. Absorbance was measured at 765 nm, and the results were expressed as equivalents of gallic acid, used as the standard (mg·eq·GA g^−1^ DW). TF were measured following the method described by Zhishen et al. [57] on the basis of the nitration of catechol groups in aromatic rings and their reaction with AlCl_3_ at an alkaline pH. Absorbance was measured at 510 nm, and the concentration of flavonoids was expressed in equivalents of the standard, catechin (mg eq C. g^−1^ DW).

### 4.9. Enzymatic Antioxidant Activities

Crude protein extracts were prepared from the leaf material, frozen, and stored at −75 °C, following the procedure described in Gil et al. [36]. The protein concentration in extracts was determined according to Bradford [58] by the Bio-Rad reagent and bovine serum albumin (BSA) as the standard. The specific activities of the four antioxidant enzymes in the protein extracts were determined by spectrophotometric assays.

Superoxide dismutase (SOD) activity was determined by monitoring spectrophotometrically at 560 nm the inhibition of nitroblue tetrazolium (NBT) photoreduction in reaction mixtures containing riboflavin as the source of superoxide radicals. One SOD unit was defined as the amount of enzyme to cause 50% inhibition of the NBT photoreduction under the assay conditions [59].

Catalase (CAT) activity was measured following the consumption of H_2_O_2_ added to the extracts by the decrease in absorbance at 240 nm. One CAT unit was defined as the amount of enzyme decomposing 1 mmol of H_2_O_2_ per minute at 25 °C [60].

Ascorbate peroxidase (APX) activity was determined following ascorbate oxidation in the presence of the plant extract by the decrease in absorbance at 290 nm. One APX unit was defined as the amount of enzyme to catalyze the consumption of 1 mmol of ascorbate per minute at 25 °C [61].

Glutathione reductase (GR) activity was quantified following the oxidation of NADPH, the cofactor in the reaction of oxidized glutathione (GSSG) reduction, by a reduction in absorbance at 340 nm. One GR unit was defined as the amount of enzyme to oxidize 1 mmol of NADPH per minute at 25 °C [62]. The minor modifications introduced into the originally published assays of CAT, SOD, and GR are described in Gil et al. [36].

### 4.10. Statistics Analysis

Data were analyzed by program SPSS v. 16 and SYSTAT v. XVI. Before the analysis of variance, the Shapiro–Wilk test was used to check for the validity of normality assumption and Levene’s test was used for the homogeneity of variance. If the ANOVA requirements were accomplished, the significance of the differences among treatments was tested by a one-way ANOVA at the 95% confidence level and post hoc comparisons were made using the Tukey HSD (honestly significant difference) test. A factorial ANOVA was performed for all parameters analyzed in the plants, considering two factors of variability—treatment and species—and their interaction. A dendrogram according to all the parameters recorded in the water-stressed plants was built by clustering the four species by the nearest neighbor method, based on squared Euclidean distances. The parameters showing a significant variation between the treatments measured in all plants (control and water-stressed) were correlated using a PCA. All the means throughout the text are followed by SE.

## 5. Conclusions

The investigated *Limonium* species showed relatively good tolerance to water deficit stress under controlled experimental conditions on the basis of some constitutive mechanisms of defense, such as the active transport of mono- and divalent ions to aerial plant parts, which can help to maintain cellular osmotic balance and avoid drought-induced leaf dehydration, or the marked activity of some antioxidant enzymes detected in the non-stressed controls. In addition, water stress-induced responses contributed to drought tolerance, including the accumulation of specific osmolytes and the activation of enzymatic antioxidant systems. Interestingly, although the four species are closely related genetically, their induced responses to water stress differed qualitatively and quantitatively with regard to the contribution of different osmolytes and enzyme activities to those tolerance mechanisms.

The behavior of *L. santapolense* under stress differed from that of the other three species, as indicated by the PCA and cluster analyses—*L. santapolense* was the species that showed the most strongly induced responses to the water stress treatment, which agrees with the fact that it naturally grows in a more arid environment than the habitats where the seeds of the other three taxa were collected. Surprisingly, however, this apparently greater efficiency in the response to water deficit did not lead to higher tolerance in our experiments. On the contrary, the determination of several growth parameters demonstrated that *L. santapolense* was the most affected species by water stress. These results strongly suggest that other mechanisms of defense that are not activated in greenhouse experiments, most likely morphological adaptations of the plants, are responsible for this species’ tolerance to drought in its natural habitat. Further studies in the field, which combine biochemical analyses of the plant material with morphological studies of plants, especially of their root system, are required to confirm this hypothesis.

## Figures and Tables

**Figure 1 plants-08-00506-f001:**
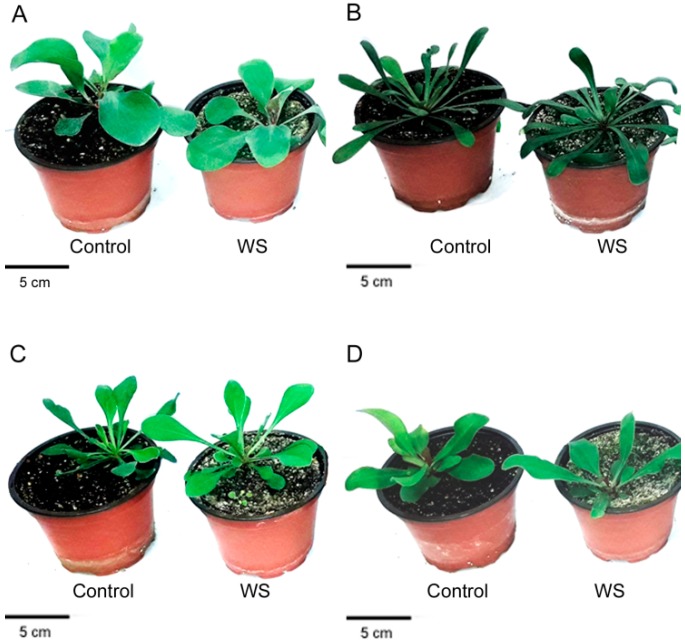
Effect of the 1 month water stress treatment (WS) in the four *Limonium* species under study: *L. santapolense* (**A**), *L. virgatum* (**B**), *L. girardianum* (**C**), and *L. narbonense* (**D**).

**Figure 2 plants-08-00506-f002:**
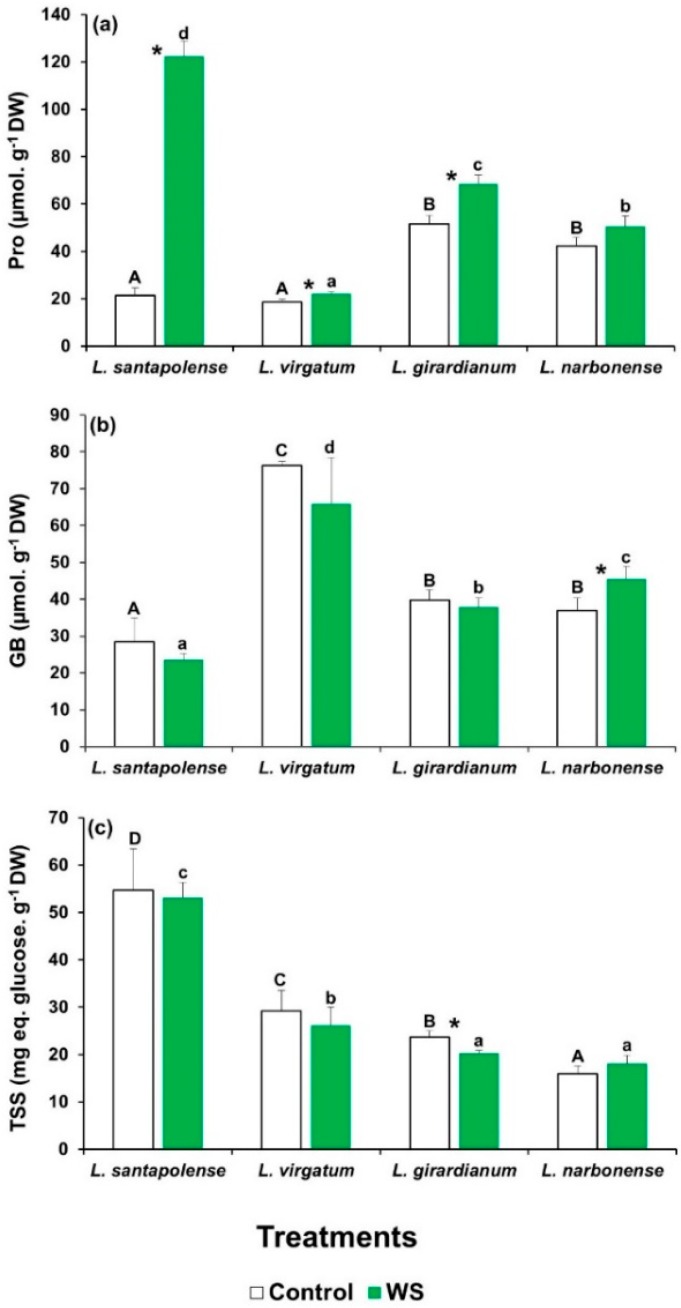
Osmolyte contents in the leaves of the four studied *Limonium* species. Proline (Pro) (**a**), glycine betaine (GB) (**b**), and total soluble sugar (TSS) levels (**c**), after 1 month of water stress treatment (WS) and in the control plants. The shown values are means with SE (*n* = 5). Different letters above the bars indicate significant differences between species, for control (capital letters) and water-stressed (lower-case letters) plants. Asterisks denote significant differences between treatments for each species, according to Tukey’s test (α = 0.05).

**Figure 3 plants-08-00506-f003:**
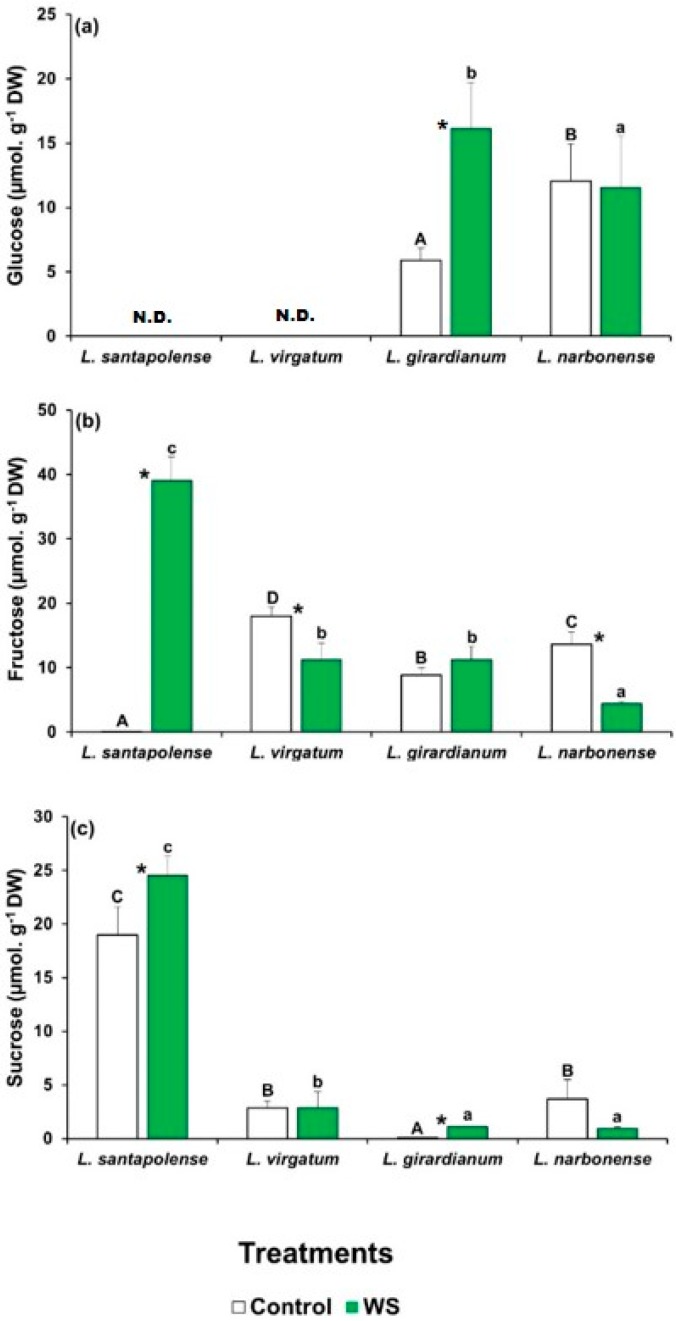
Soluble sugar contents in the leaves of the four studied *Limonium* species. Glucose (Glu) (**a**), fructose (Fru) (**b**), and sucrose (Suc) (**c**) levels after 1 month of water stress treatment (WS) and in the control plants. The shown values are means with SE (*n* = 5). Different letters above the bars indicate significant differences between species, for control (capital letters) and water-stressed (lower-case letters) plants. Asterisks denote significant differences between treatments for each species, according to Tukey’s test (α = 0.05).

**Figure 4 plants-08-00506-f004:**
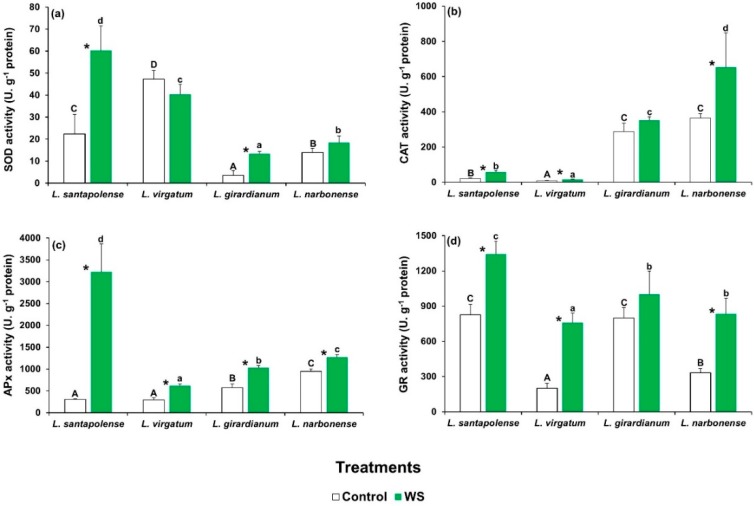
Activity of the antioxidant enzymes in the leaves of the four studied *Limonium* species after 1 month of water stress treatment (WS) and in the control plants. The graphs show the specific activities of (**a**) superoxide dismutase (SOD), (**b**) catalase (CAT), (**c**) ascorbate preoxidase (APX), and (**d**) glutathione reductase (GR) as mean values with SE (*n* = 5). Different letters above the bars indicate significant differences between species, for control (capital letters) and water-stressed (lower-case letters) plants. Asterisks denote significant differences between treatments for each species, according to the Tukey test (α = 0.05).

**Figure 5 plants-08-00506-f005:**
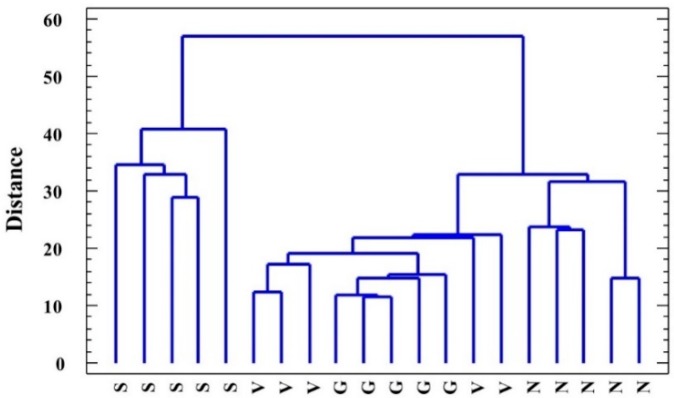
Clustering of the analyzed *Limonium* species: *L. santapolense* (S.), *L. virgatum* (V.), *L. girardianum* (G.), and *L. narbonense* (N.), by the nearest neighbor method, on the basis of squared Euclidean distances according to all the parameters registered in the water-stressed plants.

**Figure 6 plants-08-00506-f006:**
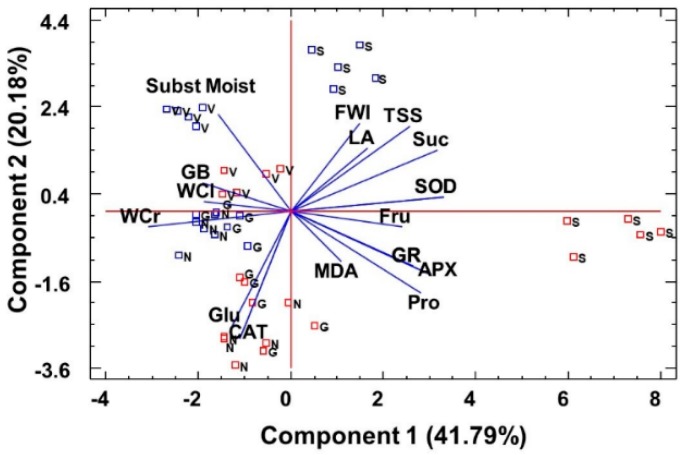
Principal component analysis (PCA). Changes in growth parameters, osmolyte levels, and antioxidant enzyme activities in the plants grown under water stress conditions for 1 month (red squares) versus the corresponding controls (blue squares); the non-stressed plants of the investigated *Limonium* species: *L. santapolense* (S.), *L. virgatum* (V.), *L. girardianum* (G.), and *L. narbonense* (N.), in correlation to substrate moisture. Each square corresponds to an individual analyzed plant. FWL, leaf fresh weight; LA, leaf surface area; WCL, water content percentage in leaves; WCR, water content percentage in roots; Pro, proline; GB, glycine betaine; TSS, total soluble sugars; Glu, glucose; Fru, fructose; Suc, sucrose; MDA, malendialdehyde; SOD, superoxide dismutase; CAT, catalase; APX, ascorbate peroxidase; GR, glutathione reductase.

**Table 1 plants-08-00506-t001:** Substrate humidity (%) of the control (C) and water-stressed (WS) *Limonium* plants grown in greenhouses. Asterisks indicate significant differences between treatments per species. Letters denote significant differences between species per treatment (capital letters for the control plants and lower-case letters for the plants subjected to 1 month water stress) at the 95% confidence level. Mean values are followed by SE (*n* = 5).

Variable	Treatment	*Limonium santapolense*	*Limonium virgatum*	*Limonium girardianum*	*Limonium narbonense*
Humidity (%)	C	45.42 ± 1.43 *^A^	45.42 ± 1.25 *^A^	45.39 ± 1.43 *^A^	43.67 ± 0.71 *^A^
	WS	10.70 ± 1.25 *^a^	14.01 ± 1.43 *^a^	12.20 ± 1.10 *^a^	12.26 ± 0.34 *^a^

**Table 2 plants-08-00506-t002:** The mean leaf fresh weight (FWL), leaf area (LA), leaf water content (WCL), root water content (WCR), and photosynthetic pigments, chlorophyll a (Chl a), chlorophyll b (Chl b), and carotenoids (Caro) levels of the control (C) and water-stressed (WS) *Limonium* plants. DW: dry weight. Asterisks indicate significant differences between treatments per species. Letters denote significant differences between species per treatment (capital letters for the control plants and lower-case letters for the plants subjected to 1 month water stress) at the 95% confidence level. Mean values are followed by SE (*n* = 5).

Variable	Treatment	*L. santapolense*	*L. virgatum*	*L. girardianum*	*L. narbonense*
FWL (g)	C	5.02 ± 0.42 *^B^	2.33 ± 0.47 ^A^	2.49 ± 0.43 ^A^	2.05 ± 0.14 *^A^
	WS	3.44 ± 0.12 *^b^	2.54 ± 0.66 ^b^	2.19 ± 0.27 ^b^	1.47 ± 0.18 *^a^
LA (cm^2^)	C	17.01 ± 1.24 *^C^	4.45 ± 0.20 ^A^	6.06 ± 0.34 ^AB^	8.60 ± 0.65 ^B^
	WS	10.26 ± 0.74 *^b^	3.87 ± 0.25 ^a^	6.27 ± 0.39 ^a^	6.87 ± 0.63 ^ab^
WCL (%)	C	84.55 ± 0.26 *^A^	87.50 ± 0.46 ^C^	86.88 ± 0.37 ^BC^	85.21 ± 0.97 ^AB^
	WS	83.61 ± 0.41 *^a^	86.44 ± 0.88 ^bc^	87.95 ± 0.40 ^c^	84.26 ± 0.76 ^ab^
WCR (%)	C	61.44 ± 5.50 ^A^	74.85 ± 1.18 ^B^	78.82 ± 1.47 ^B^	80.36 ± 1.01 *^B^
	WS	52,74 ± 5.08 ^a^	77.68 ± 1.24 ^b^	73.46 ± 1.93 ^b^	74.39 ± 0.36 *^b^
Chl a (mg g^−1^ DW)	C	2.37 ± 0.17 ^AB^	1.74 ± 0.18 ^A^	2.63 ± 0.13 ^B^	1.97 ± 0.19 ^AB^
	WS	2.27 ± 0.21 ^ab^	2.18 ± 0.21 ^ab^	2.75 ± 0.17 ^b^	1.59 ± 0.13 ^a^
Chl b (mg g^−1^ DW)	C	1.11 ± 0.01 ^A^	1.04 ± 0.06 ^A^	1.07 ± 0.10 ^A^	0.99 ± 0.14 ^A^
	WS	1.14 ± 0.25 ^a^	1.22 ± 0.18 ^a^	1.03 ± 0.06 ^a^	0.76 ± 0.09 ^a^
Caro (mg g^−1^ DW)	C	1.19 ± 0.07 ^A^	1.36 ± 0.11 ^A^	0.97 ± 0.08 ^A^	1.30 ± 0.21 ^A^
	WS	1.02 ± 0.11 ^a^	1.12 ± 0.09 ^a^	0.99 ± 0.04 ^a^	0.93 ± 0.11 ^a^

**Table 3 plants-08-00506-t003:** Mono- and divalent ion contents (µmol g^−1^ DW) and K^+^/Na^+^ ratios in the roots and leaves of the control (C) and water-stressed (WS) *Limonium* plants. Asterisks indicate significant differences between treatments per species. Letters denote significant differences between species per treatment (capital letters for the control plants and lower-case letters for the plants subjected to 1 month water stress) at the 95% confidence level. Mean values are followed by SE (*n* = 5).

Ion	Treatment	*L. santapolense*	*L. virgatum*	*L. girardianum*	*L. narbonense*
Na^+^ _roots_	C	137.22 ± 23.17 *^A^	127.99 ± 2.19 ^A^	119.52 ± 1.68 *^A^	174.17 ± 16.09 *^A^
	WS	209.96 ± 42.13 *^ab^	116.44 ± 17.76 ^a^	184.18 ± 11.05 *^a^	238.37 ± 22.70 *^b^
Na^+^ _leaves_	C	450.01 ± 10.61 ^A^	473.66 ± 27.32 ^A^	534.93 ± 29.99 ^A^	552.11 ± 67.78 ^A^
	WS	499.65 ± 25.75 ^b^	426.83 ± 7.46 ^a^	513.47 ± 22.41 ^b^	416.29 ± 18.90 ^a^
K^+^ _roots_	C	279.85 ± 35.35 ^A^	353.33 ± 8.86 ^AB^	278.78 ± 2.46 ^A^	382.72 ± 32.19 ^B^
	WS	242.79 ± 35.36 ^a^	300.29 ± 42.01 ^ab^	326.49 ± 22.23 ^ab^	406.74 ± 36.69 ^b^
K^+^ _leaves_	C	833.33 ± 20.98 ^A^	977.25 ± 43.11 ^A^	974.47 ± 32.05 ^A^	981.48 ± 97.45 ^A^
	WS	839.27 ± 27.83 ^ab^	897.79 ± 15.08 ^ab^	1031.71 ± 75.86 ^b^	823.25 ± 60.43 ^a^
K^+^/Na^+^ _roots_	C	2.08 ± 0.18 *^A^	2.75 ± 0.01 ^C^	2.33 ± 0.05 *^B^	2.20 ± 0.02 *^AB^
	WS	1.30 ± 0.16 *^a^	2.60 ± 0.07 ^c^	1.77 ± 0.03 *^b^	1.72 ± 0.09 *^b^
K^+^/Na^+^ _leaves_	C	1.85 ± 0.04 ^A^	2.08 ± 0.26 ^A^	1.83 ± 0.04 ^A^	1.80 ± 0.09 ^A^
	WS	1.69 ± 0.06 ^a^	2.11 ± 0.15 ^b^	1.99 ± 0.06 ^b^	1.96 ± 0.06 ^b^
Cl^−^ _roots_	C	186.77 ± 18.28 *^A^	236.95 ± 5.35 ^A^	203.10 ± 28.32 *^A^	355.43 ± 35.59 ^B^
	WS	203.10 ± 51.70 *^a^	228.49 ± 17.34 ^a^	299.01 ± 20.26 *^ab^	456.98 ± 41.80 ^b^
Cl^−^ _leaves_	C	727.78 ± 51.33 ^A^	787.02 ± 75.02 ^A^	767.27 ± 32.36 ^A^	971.50 ± 79.81 ^A^
	WS	836.10 ± 47.45 ^a^	892.80 ± 34.84 ^a^	856.27 ± 17.34 ^a^	999.35 ± 82.48 ^a^
Ca^2+^ _roots_	C	13.29 ± 2.03 *^B^	4.48 ± 0.64 *^A^	6.18 ± 1.20 *^A^	14.23 ± 0.94 *^B^
	WS	6.18 ± 2.20 *^a^	10.35 ± 1.14 *^a^	9.69 ± 0.66 *^a^	24.28 ± 2.67 *^b^
Ca^2+^ _leaves_	C	84.88 ± 7.03 ^A^	66.74 ± 10.43 ^A^	61.93 ± 12.40 ^A^	53.01 ± 12.13 ^A^
	WS	97.12 ± 5.66 ^bc^	68.46 ± 14.90 ^ab^	117.57 ± 20.63 ^b^	37.20 ± 1.70 ^b^
Mg^2+^ _roots_	C	64.24 ± 3.36 *^C^	40.05 ± 0.36 *^A^	50.93 ± 3.01 ^B^	76.33 ± 3.51 ^D^
	WS	50.93 ± 5.51 *^a^	57.21 ± 6.28 *^a^	64.07 ± 5.07 ^ab^	78.34 ± 4.52 ^b^
Mg^2+^ _leaves_	C	456.63 ± 29.80 ^A^	401.16 ± 40.36 ^A^	486.11 ± 61.29 ^A^	320.88 ± 34.63 ^A^
	WS	538.36 ± 22.53 ^c^	429.16 ± 49.70 ^b^	555.07 ± 13.81 ^c^	260.80 ± 38.60 ^a^

**Table 4 plants-08-00506-t004:** Malondialdehyde (MDA) and hydrogen peroxide (H_2_O_2_) concentrations, α,α-diphenyl-β-picrylhydrazyl (DPPH) free radical scavenging activity, total phenolic compounds (TPC, expressed as mg equivalents of gallic acid), and total flavonoid (TF, expressed as mg equivalents of catechin) contents in the leaf extracts from the plants of the four selected *Limonium* species. Asterisks indicate significant differences between treatments per species. Different letters (capital for the non-stressed controls and lowercase for the plants subjected to 1 month of water deficit stress) indicate significant differences between species per treatment at the 95% confidence level. Mean values are followed by SE (*n* = 5).

Variable	Treatment	*L. santapolense*	*L. virgatum*	*L. girardianum*	*L. narbonense*
MDA (nmol g^−1^ DW)	C	103.58 ± 16.92 ^A^	83.93 ± 18.61 ^A^	149.16 ± 16.74 ^A^	80.09 ± 7.60 *^A^
	WS	152.59 ± 28.69 ^a^	138.82 ± 21.224 ^a^	162.53 ± 17.85 ^a^	135.28 ± 14.73 *^a^
H_2_ O_2_ (µmol g^−1^ DW)	C	17.93 ± 3.29 ^A^	17.21 ± 2.26 ^A^	20.86 ± 3.49 ^A^	26.68 ± 3.24 ^A^
	WS	25.25 ± 3.80 ^b^	12.89 ± 1.28 ^a^	18.59 ± 0.38 ^ab^	21.94 ± 1.01 ^b^
DPPH (%)	C	84.72 ± 6.10 ^C^	76.94 ± 0.99 ^BC^	15.28 ± 2.97 ^A^	52.34 ± 11.27 ^B^
	WS	82.60 ± 9.86 ^b^	62.17 ± 6.90 ^ab^	34.13 ± 7.97 ^a^	41.81 ± 5.70 ^ab^
TPC (mg eq. GA g^−1^ DW)	C	24.41 ± 4.18 ^B^	11.50 ± 6.15 ^A^	6.40 ± 1.31 ^A^	12.57 ± 3.01 ^A^
	WS	23.96 ± 1.81 ^b^	6.15 ± 0.40 ^a^	6.56 ± 1.18 ^a^	9.25 ± 2.77 ^a^
TF (mg eq. C g^−1^ DW)	C	1.96 ± 0.37 ^AB^	1.26 ± 0.16 ^A^	0.71 ± 0.13 ^A^	2.95 ± 0.61 ^B^
	WS	1.80 ± 0.28 ^a^	1.00 ± 0.13 ^a^	1.22 ± 0.18 ^a^	1.90 ± 0.45 ^a^

**Table 5 plants-08-00506-t005:** Factorial ANOVA (*F* values) considering the effect of treatment (T), species (S) and their interactions (T × S) on the measured variables. FW: fresh weight; LA: leaf area; WC_l_: water content of leaves; WC_r_: water content of roots; Chl a: chlorophyll a; Chl b: chlorophyll b; Caro: carotenoids; Pro: proline, GB: glycine betaine; TSS: total soluble sugars; Glu: glucose, Fru: fructose: Suc: sucrose; MDA: malondialdehyde; TPC: total phenolic compounds; TF: total flavonoids; SOD, superoxide dismutase; CAT, catalase; APX, ascorbate peroxidase, GR glutathione reductase. *, **, *** significant at *p* = 0.05, 0.01, and 0.001, respectively.

Growth	**Parameter**	**Treatment (T)**	**Species (S)**	**Interaction (T × S)**
	FW_l_	2.88	10.83 ***	1.36
	LA	11.59 **	39.65 ***	2.96 *
	WC_l_	1.23	14.81 ***	1.45
	WC_r_	14.16 ***	30.38 ***	4.11 *
Photosynthtetic pigments	Chl a	0.01	10.87 ***	2.05
	Chl b	0.1	1.91	0.66
	Caro	5.97 *	1.77	1.06
Mono and divalent ions	Na^+^_r_	10.87 **	6.06 **	1.89
	Na^+^_l_	3.01	1.91	2.96 *
	K^+^_r_	1.37	133.84 ***	6.09 **
	K^+^_l_	1.31	3.33 *	1.55
	Cl^−^_r_	8.68 ***	14.07 ***	2.90 *
	Cl^−^_l_	5.45 *	3.30 *	1.33
	Ca^2+^_r_	0.05	7.39 ***	1.37
	Ca^2+^_l_	3.25	9.44 ****	2.80
	Mg^2+^_r_	3.29	19.11 ***	6.44 **
	Mg^2+^_l_	2.08	14.55 ***	1.48
Compatible solutes	Pro	172.75 ***	80.14 ***	86.56 ***
	GB	1.68	22.21 ***	2.81
	TSS	0.27	37.72 ***	0.23
	Glu	5.51 *	39.60 ***	6.09**
	Fru	25.67 ***	14.65 ***	79.13 ***
	Suc	5.35 *	3.27 *	1.15
Oxidative stress markers and antioxidants	MDA	9.26 **	2.40	0.50
	H_2_O_2_	0.29	4.33 *	2.29
Antioxidants	TPC	2.23	27.97 ***	0.73
	TF	1.05	10.06 **	2.21
	SOD	10.84 **	28.29 ***	10.04 ***
	CAT	12.63 *	73.30 ***	5.39 **
	APX	71.35 ***	21.84 ***	28.95 ***
	GR	42.53 ***	16.83 ***	1.47

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
