# Peer review of "Qualitative and Quantitative Differences in Osmolytes Accumulation and Antioxidant Activities in Response to Water Deficit in Four Mediterranean Limonium Species"

_plants, 2019, doi:10.3390/plants8110506_

Round 1

Reviewer 1 Report

Comments

The explanation of Figure 5 is not clear. For instance, there were five S, V, G and N, respectively. I can see each label indicated a species, but it need to be written clearly. And why is there five for each species, which also need to explain. So the author need to explain more about the X-axis. Line 279, please indicate the two main principle components that explained 63% of the total variability. Explain the difference between the red and black square in Figure 6. Which one is stress or control? The number of the reference need to reduce. 93 for the research paper is too many. Analyzed the data from Table 1, 2 3 using the method of multi-factor analysis of variance. Then, we can see the interaction between the factors. Line 57-58. They include minimising stomatal and cuticular water loss, 57 increased root growth for optimising water uptake from the soil, or by the synthesis of compatible 58 solutes, such as proline, glycine betaine or soluble carbohydrates, among others [12–18]. This is not closely related. The parameters that you mentioned concerning with drought tolerance should be those that you are going to test so that you can have more discussion later on. One of the reason that you have so many references could be that you cited too many things that were not that related. Try to delete those information and extract the essence. Line 73-75. The genus Limonium (Plumbaginaceae family), comprises more than 400 species, is well represented in the Mediterranean and includes numerous endemics in the area of the present study [30]. So why do you choose these four species in the current study? Line 75-76. Limonium species are well documented regarding their responses to salt stress as this is an 75 emblematic genus of halophytes with salt excretory glands [31–37]. I suggested that you removed some sentences about salt stress, since your focus is drought stress and you already have too many references and long papers. Try to delete the redundant information. Line 78-83. However, salinity is not the only constraint for plants in salt marshes, as many other stressful factors frequently occur simultaneously. This situation is especially evident in this type of habitat, where considerable seasonal changes in environmental conditions are detected, in addition to the changes in soil salinity. In the Mediterranean basin, soil salinity shows a notable increase in summer [38], accompanied by a drastic reduction in soil water content, due to intense evapotranspiration and lack of rainfall. The author should focus on water deficit. Line 84- It is too much information with few closely related. Line 100- The author spend too much efforts to show the previous work that is related to temperature and salt stress. Instead, the author should move to water deficit.   Why the stress period last one month? What make the author choose to treat for one month? How many plants do the author treat for control and stress? Five is not that enough. The author mentioned that ‘ Five leaves from each plant were selected randomly’. But upper or bottom leaf? Old or young leaf? And the author measured FWL, WCL and LA. Meanwhile, the pigment, ion content, enzyme activity and so on were also detected. So were they from the same or different leaves from the same plants or even different plants? Explain why there is no glucose detected in S and V species. Was it reported in previous papers? Line 324-326. ‘however, contrary to this general trend, a recent analysis on 13 Limonium species from Balearic Islands showed that larger leaf size was correlated to increased growth capacity and water use efficiency [43].’ Give the reason why the previous author conclude like that. And discuss it with your result and make your own conclusion. Line 352-366 with no reference, in the discussion, you need to have more citation to compare with your results. The same problem for Line 421-433 The full name should be given in the first time when it was mentioned. Then just use the abbreviation. For instance, Malondialdehyde (MDA) in Line 395 should be just MDA. Check this for other abbreviations. Generally, the section of introduction and discussion should be re-written to focus on the topic. Especially for the section of discussion, the author should discuss by combining with recent papers in depth.

Author Response

We would like to thank the two reviewers for their comments and suggestions on the original version, which have allowed us improving the quality of the manuscript. We generally agree with all their comments, and have modified the manuscript accordingly. Detailed, point-by-point answers to the reviewers’ comments and concerns are given below. We believe that we have satisfactorily answered all questions raised by the reviewers, and hope that they, and the editor, consider the revised version acceptable for publication in Plants.

Reviewer report 1

The explanation of Figure 5 is not clear. For instance, there were five S, V, G and N, respectively. I can see each label indicated a species, but it need to be written clearly. And why is there five for each species, which also need to explain. So the author need to explain more about the X-axis.

The reviewer is right. Indeed, the legend of Fig. 5 was not complete, as we did not include some details, such as the abbreviations used for each species (S, V, G and N), or that the cluster analysis was performed using all variables measured in the water-stressed plants, including growth parameters. Although this information is mentioned in the text, in the revised version it has been included also in the legend, as we agree that the figures should be understandable without reference to the text.

Line 279, please indicate the two main principle components that explained 63% of the total variability. Explain the difference between the red and black square in Figure 6. Which one is stress or control?

The first component, which explains the highest variability of the data (41.79%), is related to the treatment, negatively to the substrate moisture and positively to several osmolytes (Pro, TSS, Fru, and Suc) and antioxidant enzymes (APX; GR, SOD). The second component, explaining an additional 20%, is more related to the type of species, separating L. santapolense and L. virgatum from L. narbonense and L. giradianum. Each square in the figure represents one single analysed individual, and we used the red colour for the plants subjected to the water stress treatment and blue colour (it is not black, although it may not be clear) for non-stressed, control plants. This information has been added to the text in the Results Section, and to the legend of Fig. 6 in the revised version of the manuscript.

The number of the reference need to reduce. 93 for the research paper is too many.

Thank you for your suggestion, we did include too many references. We have reduced this number from 93 to 74, eliminating non-essential or redundant references. A more drastic reduction would have implied the deletion of some relevant citations, considering that 19 references were necessarily included in the Material and Methods section, as we performed many different types of biochemical analysis. Considering this, we think that the number of references is now appropriate for a research paper.

Analyze the data from Table 1, 2 3 using the method of multi-factor analysis of variance. Then, we can see the interaction between the factors.

A multi-factor analysis of variance has been performed and is referred to in the revised version of the manuscript, in the Methods, results and Discussion sections. The results of this analysis are shown in Table 5.

Line 57-58. They include minimising stomatal and cuticular water loss, 57 increased root growth for optimising water uptake from the soil, or by the synthesis of compatible 58 solutes, such as proline, glycine betaine or soluble carbohydrates, among others [12–18]. This is not closely related. The parameters that you mentioned concerning with drought tolerance should be those that you are going to test so that you can have more discussion later on. One of the reason that you have so many references could be that you cited too many things that were not that related. Try to delete those information and extract the essence. 

Several sections of the manuscript have been extensively modified, partly to improve the English language of the manuscript, but at the same time deleting some non-essential information and therefore reducing the number of references, as suggested by the reviewer.

Line 73-75. The genus Limonium (Plumbaginaceae family), comprises more than 400 species, is well represented in the Mediterranean and includes numerous endemics in the area of the present study [30]. So why do you choose these four species in the current study? 

This work was partially financed by a research project dealing with endemic halophytes from the area of Valencia. Two of the species selected for this study (Limonium santapolense and L. girardianum) have high conservation value, as they are endemic and highly threatened taxa, the first restricted to a few localities in the Valencian Community and the second in SE Spain and S. France, where it is strictly protected. The other two species have a broader distribution throughout the Mediterranean but are becoming rare in our region due to loss of their habitats. Therefore all four are considered as of special interest, and have not been investigated by other groups. Also, the same four species have been the target of previous studies of our group (some of them in collaboration with other groups), focused mostly on their responses to salt stress during seed germination and vegetative growth.

Line 75-76. Limonium species are well documented regarding their responses to salt stress as this is an 75 emblematic genus of halophytes with salt excretory glands [31–37]. I suggested that you removed some sentences about salt stress, since your focus is drought stress and you already have too many references and long papers. Try to delete the redundant information

This paragraph was deleted, following the reviewer’s suggestion.

Line 78-83. However, salinity is not the only constraint for plants in salt marshes, as many other stressful factors frequently occur simultaneously. This situation is especially evident in this type of habitat, where considerable seasonal changes in environmental conditions are detected, in addition to the changes in soil salinity. In the Mediterranean basin, soil salinity shows a notable increase in summer [38], accompanied by a drastic reduction in soil water content, due to intense evapotranspiration and lack of rainfall. The author should focus on water deficit

Again, we absolutely agree with the reviewer. We have deleted or reduced comments on salt stress responses throughout the text, which may alter the focus of the manuscript not being directly related to the present study.

Line 84- It is too much information with few closely related. 

We agree; this part of the text was too descriptive and not relevant for our study. General information on the taxa under study is considerably more concise in the revised version of the manuscript.

Line 100- The author spend too much efforts to show the previous work that is related to temperature and salt stress. Instead, the author should move to water deficit.

True; we have substantially reduced the references to our previous results on the effects of temperature and salt stress on the selected Limonium species.

  Why the stress period last one month? What make the author choose to treat for one month?

As we explained before, this study is part of a research project on responses of endemic halophytes from the region of Valencia, SE Spain, to abiotic stress. We have tried to standardise, as much as possible, the protocols and conditions of the treatments in the greenhouse, including the time the plants are subjected to stress. In this way, we may be able to compare the results obtained with taxa from different groups – unless, obviously, there are big differences in their relative stress tolerance. Based on our previous work, one-month treatment appears to be a convenient period, as the effect of stress is noticeable on plant growth and changes in several biochemical stress markers, but it is not extreme; the plants survive, do not show strong symptoms of wilting or senescence and sufficient material can be collected for biochemical analysis after the stress treatment.

How many plants do the author treat for control and stress? Five is not that enough. The author mentioned that ‘ 

The number of plants per treatment and per species is 5. We agree that this is a rather low number when dealing with crops or other type of easily available plant material, but it is usual when analysing wild species, and especially endemic plants. In such situations, the number of seeds is limited, as they are generally collected in the wild (as in this case), often from protected areas, and germination is quite often difficult. Therefore, many articles dealing with wild or endemic taxa report work performed on five (sometimes even less) plants per treatment and species. For example, in the last few years we have published several papers using this number of plants (n = 5) in well-known journals in the field, such as those indicated bellow:

- Al Hassan M. et al. Native-invasive plants vs. halophytes: stress tolerance mechanisms in two related species. Front. Plant Sci. 7, 473, 10.3389/fpls.2016.00473 (2016).

- Al Hassan et al. Stress tolerance mechanisms in Juncus: responses to salinity and drought in three Juncus species adapted to different natural environments. Funct. Plant Biol. 43, 949–960 (2016).

- Al Hassan M. et al. Effects of salt stress on three ecologically distinct Plantago species. PLoS ONE 11, e0160236, 10.1371/journal.pone.0160236 (2016)

- Al Hassan et al. Antioxidant responses under salinity and drought in three closely related wild monocots with different ecological optima. AoB Plants 9, plx009, 10.1093/aobpla/plx009 (2017)

- Al Hassan et al. Unraveling salt tolerance mechanisms in halophytes: A comparative study on four Mediterranean Limonium species with different geographic distribution patterns. Front. Plant Sci. 8, 1438, 10.3389/fpls.2017.01438 (2017)

- Pardo-Domènech et al. Proline and glycine betaine accumulation in two succulent halophytes under natural and experimental conditions. Plant Biosyst. 150, 904–915 (2016).

            Still in relation to this issue, we would like to mention that, as shown by the PCA in Fig. 6, the five individuals analysed for each treatment, belonging to the same species, are grouped together and not scattered throughout the graph. This clearly indicates the homogeneity and reliability of the data obtained.

Five leaves from each plant were selected randomly’. But upper or bottom leaf? Old or young leaf?

Regarding the type of leaves, the four species have rosette leaves, so all are basal. For the different biochemical analysis were used all leaves sampled; five from each plant were selected only for measuring leaf area. All leaves used this measurement were young. 

And the author measured FWL, WCL and LA. Meanwhile, the pigment, ion content, enzyme activity and so on were also detected. So were they from the same or different leaves from the same plants or even different plants?

As we explained above, we pooled all leaves produced by each plant, to be used in the different biochemical assays. At the end of the treatments, all leaves and all roots were separately sampled from each individual. Part of the material was stored -75ºC and the rest was dried; dry plant material was mostly used for ion contents measurements, and to calculate dry weights and water content percentages.

Explain why there is no glucose detected in S and V species. Was it reported in previous papers?

There is a large variability regarding reports on glucose levels in plants of this genus. Glucose was reported in some Limonium species (Liu & Grieve, 2009; Gagneul et al. 2007). In some species a great seasonal variation was detected (Furtana et al. 2013). In a previous study that we performed on the effect of salt stress on the four species (Al Hassan et al. 2017), levels of glucose were low and did not show variation under stress. In the present study glucose levels were too low to be detected by HPLC analysis in two species (L. santapolense and L. virgatum). As such we can say the glucose do not play a considerable role in osmoregulation.

Liu, X. & Grieve, C. Accumulation of chiro-inositol and other nonstructural carbohydrates in Limonium species in response to saline irrigation waters. J. Am. Soc. Hortic. Sci. 134, 329–336 (2009);

Furtana, G. B., Dumani, H. & Tipirdamaz, R. Seasonal changes of inorganic and organic osmolyte content in three endemic Limonium species of Lake Tuz (Turkey). Turk. J. Bot. 37, 455–463 (2013).

Gagneul, D.; Aïnouche, A.; Duhazé, C.; Lugan, R.; Larher, F.R.; Bouchereau, A. A. reassessment of the function of the so-called compatible solutes in the halophytic Plumbaginaceae Limonium latifolium. Plant Physiol. 2007, 144, 1598–1611.

Al Hassan et al. Unraveling salt tolerance mechanisms in halophytes: A comparative study on four Mediterranean Limonium species with different geographic distribution patterns. Front. Plant Sci. 8, 1438, 10.3389/fpls.2017.01438 (2017)

Line 324-326. ‘however, contrary to this general trend, a recent analysis on 13 Limonium species from Balearic Islands showed that larger leaf size was correlated to increased growth capacity and water use efficiency [43].’ Give the reason why the previous author conclude like that. And discuss it with your result and make your own conclusion.

We included this citation in the original version of the manuscript even though the results are rather confusing and is apparently in contrast to our findings. The study on Limonium species from Balearic Islands is based on a comparison of net assimilation rate (NAR) with leaf area ratio (LAR). The main conclusion is that NAR is a more relevant parameter, as the species with larger leaves had higher water consumption but also higher water use efficiency. As photosynthesis was not analysed in our study, those data are not relevant for our work and we now consider that it does not make much sense to include this citation (as the reviewer previously indicated for other citations not directly related with the work described here). Therefore, to avoid confusing the reader with long explanations not related to the present study, we have deleted this reference from the revised version. As a higher leaf area is generally associated with a higher surface of evapotranspiration and therefore accentuated water loss, we introduced another reference in agreement with our findings. The paragraph is now more coherent.

Line 352-366 with no reference, in the discussion, you need to have more citation to compare with your results.

Our results were compared with other data, and references were added

The same problem for Line 421-433

Also in this case, new references were introduced to reinforce our findings.

The full name should be given in the first time when it was mentioned. Then just use the abbreviation. For instance, Malondialdehyde (MDA) in Line 395 should be just MDA. Check this for other abbreviations.

Done

Generally, the section of introduction and discussion should be re-written to focus on the topic. Especially for the section of discussion, the author should discuss by combining with recent papers in depth

Both sections were revised, extensive paragraphs regarding other type of stress (mostly saline) were removed in order to maintain the focus on the responses to water stress in the four species. Several new references were included. The statistical analysis was included in the discussion.

Reviewer 2 Report

In this MS author study Qualitative and quantitative differences in osmolytes accumulation and antioxidant activities in response to water deficit in four Mediterranean Limonium species. MS is loosely written and need serious English edit. I have detected plagiarized stuff in it. Please take care of it.

L33, L407, L414 defence to defense

L44 affects to affect

L50, L108, L153, L267, L447-L449, L456, 463, L484, L487-L489, L491, L496-L501, L544-L546, L561-L566 This line is plagiarized

L58 optimising to optimizing

L124, L215, L496 analysed to analyzed

L138 plants was to plants were

L57, L267, L429, L591 Rephrase this line

L321 The, apparently, looks odd

L390 that, to that

L391 plants, to plants

L415 increases to increased

L418 by accumulation to by the accumulation

L420 as cofactor to as a cofactor

L478 analyser to analyzer

L502 run during to run for

L553 catalysing to catalyzing

Author Response

We would like to thank the two reviewers for their comments and suggestions on the original version, which have allowed us improving the quality of the manuscript. We generally agree with all their comments, and have modified the manuscript accordingly. Detailed, point-by-point answers to the reviewers’ comments and concerns are given below. We believe that we have satisfactorily answered all questions raised by the reviewers, and hope that they, and the editor, consider the revised version acceptable for publication in Plants.

In this MS author study Qualitative and quantitative differences in osmolytes accumulation and antioxidant activities in response to water deficit in four Mediterranean Limonium species. MS is loosely written and need serious English edit. I have detected plagiarized stuff in it. Please take care of it.

All comments and suggestions of the reviewer have been taken into account when preparing the revised version of the manuscript. The grammar and style of the text has been checked and extensively modified by a native speaker of a professional service for translation and editing of scientific texts in English. It is true that there are coincidences in the text with some of our previous publications, which we can consider as ‘unconscious’ self-plagiarism (after years working on the same general topic, we tend to use the same or similar sentences when writing background information in the Introduction or describing and discussing the results of the experiments). The wording of those fragments of the text has been modified, maintaining its intended meaning.

Regarding the minor modifications suggested by the reviewer, which we are not repeating here, all have been considered when preparing the revised version of the manuscript, in which we have introduced the requested changes. We would like to thank the reviewer for his/her careful and detailed check of the manuscript.

We have also modified the spelling of the text; the original version was written in British English and the revised version using American spelling, changing ‘analysed’ to ‘analyzed’, ‘optimising’ to ‘optimizing’, ‘catalysing to catalyzing’, etc., following the reviewer’s indications.

Round 2

Reviewer 1 Report

Generally, the author made lots of efforts to improve. But there are minor problems that can be improved. 

1.Line 576. 48 should be [48]

2. Double years in Line 731. (1996-2018)

3. For a research paper, I would say 74 references were still too much. And remove the No.75 in the end if you do not have more reference. 

Author Response

Thank you very much for your detailed observations. We reduced the number of references as you suggested. We hope that the number of 61 references in the new version of the manuscript is appropriate for a research article.

Reviewer 2 Report

I am happy with the current edited MS and It can be accepted.

Author Response

Thank you very much for your appreciation